# A Polyvocal and Contextualised Semantic Web⋆

Marieke van Erp[1][0000−0001−9195−8203] and Victor de Boer[2][0000−0001−9079−039X]

[1] KNAW Humanities Cluster
Amsterdam, the Netherlands
marieke.van.erp@dh.huc.knaw.nl
https://mariekevanerp.com
[2] Vrije Universiteit Amsterdam
Amsterdam, the Netherlands
v.de.boer@vu.nl

**Abstract.** Current AI technologies and data representations often reflect the popular or majority vote. This is an inherent artefact of the frequency bias of many statistical analysis methods that are used to create for example knowledge graphs, resulting in simplified representations of the world in which diverse perspectives are underrepresented. With the use of AI-infused tools ever increasing, as well as the diverse audiences using these tools, this bias needs to be addressed in both the algorithms analysing data, as well as in the resulting representations. In this *problems to solve before you die* submission, we explain the implications of the lack of polyvocality and contextual knowledge in the semantic web. We identify three challenges for the Semantic Web community on dealing with various voices and perspectives as well as our vision for addressing it.

**Keywords:** Culturally aware AI · Polyvocality · Contextualisation · Bias

## 1 Introduction

Biases in data can be both explicit and implicit. Explicitly, 'The Dutch Seventeenth Century' and 'The Dutch Golden Age' are pseudo-synonymous and refer to a particular era of Dutch history. Implicitly, the 'Golden Age' moniker is contested due to the fact that the geopolitical and economic expansion came at a great cost, such as the slave trade. A simple two-word phrase can carry strong contestations, and entire research fields, such as post-colonial studies, are devoted to them. However, these sometimes subtle (and sometimes not so subtle) differences in voice are as yet not often found in knowledge graphs.

One of the reasons is that much of the knowledge found in knowledge graphs is mined automatically and current AI technologies (and their ensuing data representations) often reflect the popular or majority vote. This is an inherent artefact of the frequency bias of many statistical analysis methods that are used

---

⋆ Supported by cultural-ai.nl

to create for example knowledge graphs, resulting in simplified representations of the world in which diverse perspectives are underrepresented. With the use of AI-infused tools ever increasing, as well as the diverse audiences using these tools, this bias needs to be addressed in both the algorithms analysing data, as well as in the resulting representations.

Conversations around data bias and polyvocality are taking place in for example the cultural heritage domain (cf. [15, 14]) and computational linguistics communities (cf. [8, 11]), but do not yet seem mainstream in the Semantic Web discourse. In 2010, Hendler and Berners-Lee already recognised that current knowledge mining mechanisms can bias results and recommend 'Making the different ontological commitments of competing interpretations explicit, and linked together', as this 'can permit different views of data to be simultaneously developed and explored.' [9]. Veltman noted in her 2004 paper that '[Dynamic knowledge will] allow us to trace changes of interpretation over time, have new insights and help us to discover new patterns in knowledge' [20]. Context was also flagged as the next frontier in knowledge representation at the 'Knowledge graphs: New directions for knowledge representation on the semantic web' Dagstuhl seminar in 2018 [5]. While it is known that large knowledge graphs are not always balanced (cf. [12]), that links between linked data resources contain biases (cf. [19]), and that research based on such resources might favour a Western perspective (cf. [6]), the creation of contextualised and polyvocal knowledge graphs has only gained modest traction.

As data-driven applications are permeating everyday life, it is necessary that these applications can serve as many and as diverse audiences as possible. A Dutch visitor to the Rijksmuseum in Amsterdam could recognise the Christian context of a painting depicting Saint Christopher even if s/he does not know that he is the patron saint of travellers. For a Japanese visitor, this might not be immediately apparent, and s/he needs more context to connect this to her/his cultural context, to for example recognise the parallels to Jizō, protector of travellers in some Buddhist traditions. For most other applications (e.g. in the domains of food, health, policy and transportation) a greater awareness of cultural and contextual frameworks of users would be crucial to increase engagement and understanding. Thus far, one of the ways machine learning driven approaches have dealt with this, is to add more data, at the risk of flattening nuances. At times, this has led to painful situations, for example when a Twitter trained chatbot started publishing racist, sexist and anti-semitic Tweets within a day of being released [23]. We argue that addressing this by more intelligently organised datasets to identify bias, and multiple perpectives and contexts is needed. Many of the puzzle pieces for a polyvocal and contextualised Semantic Web are already in place. The different local Wikipedias for example present different perspectives [4, 24] with varying levels of in- and between group biases [1] but this is often still expressed implicitly.

In this *problems to solve before you die* submission, we explain the implications of the lack of polyvocality and contextual knowledge in the semantic web, as well as our vision for addressing it. The examples we use come from cultural

heritage datasets as we work with those and some of the bias issues are amplified in there due to their longue durée,[3] but these issues and our proposed approach also apply to other domains.

## 2    What is a voice?

Data is not objective, but rather created from a particular perspective or view, representing a *voice*. These perspectives can be informed by cultural, historical or social conventions or a combination of those. Often, these different perspectives are contiguous, rather than disjoint. A cultural view on for example the 'The Dutch Golden Age' is that it was an era in which the Dutch economy and scientific advancements were among the most acclaimed in the world, laying the foundations for the first global multinational corporation and shaping Dutch architecture. A historical view could be that it was an era of unbridled opportunity and wealth for many in the Dutch Republic. A social view could focus on the pride associated with the achievements of this era. However, none of these views tells the entire story. The Dutch Republic maintained several colonies in Africa and Asia and was heavily involved in slave trading; the other side of the coin of wealth and pride.

These different views often come to us through objects in cultural heritage collections which can often be different things throughout their life. For example, many cult figures and other objects were taken and brought back home by missionaries -in some cases to learn about 'the other' from a euro-centric view. Such items were often (dis)qualified as "fetishes" or "idols" and Christian converts were expected to stay away converted from them. Later, these objects were removed from their original ritual settings and became part of ethnographic collections in missionary exhibitions or were sold as works of art. These reframings represent new perspectives on the same object and depend on time, culture and the object provenance. Data models that are perspective-aware should be able to trace the various changes that objects undergo in their trajectory from their original uses in indigenous religious practices into museum collections[13].

A polyvocal Semantic Web provides opportunities, models and tools to identify, represent and show users different perspectives on an event, organisation, opinion or object. Furthermore, identified individual perspectives need to be connected and clustered to be lifted from an individual perspective to a (representative) voice of a group. Identifying, representing and using such groups with fuzzy boundaries that change over time is one of the core challenges.

## 3    How can the Semantic Web deal with multiple perspectives and interpretations?

We identify three main challenges to support polyvocality on the Semantic Web: identification, representation and usage. We describe these challenges below,

---

[3] Some collections we work with have been gathered over a span of over 200 years, and the objects they describe go back even longer.

along with the current state of the art and promising research directions for addressing them.

### 3.1   Identifying polyvocality in data sources

Data sources of various modalities can represent a singular 'voice', and provide a specific perspective on the world. However, within and across data sources, there can also be multiple voices present. Moreover, when aggregating datasets, such voices can become lost in the process. The challenge here is to identify such voices. For natural language data sources, as well as for methods that extract information from structured sources, this requires Information Extraction methods that are 'bias' or 'voice'-aware. In cases where datasets are the result of aggregations, such methods should be able to (re-)identify the separate voices in the combined source.

In cases where human data providers are involved, the voices of the individual users should be maintained. These persons can also represent cultural or societal groups and this information should be available for subsequent representations. As an example, consider a crowdsourcing effort to annotate a Polynesian object in a European museum. Here, annotations provided by the European public on the one hand and annotations provided by members of the source community on the other hand will represent various perspectives. Methods that aggregate such annotations should retain these voices. An example of a method that does this for individual annotators is CrowdTruth[2], which maintains so-called 'disagreement' between annotators. Such methods can be expanded to retain this disagreement to the extent to which it represents a 'voice'.

What holds for these non-professional annotators, also holds for professionals involved in metadating or interpreting information. Different perspectives provided by such professionals should also be maintained in the information processing pipeline. Again, in the cultural heritage domain, with movements towards a diversification of the museum professionals gaining traction [22, 18], the different perspectives that such professionals provide should be maintained in the object metadata.

### 3.2   Representation of polyvocality: datamodels and formalisms

Identifying, extracting and retaining polyvocal information in data sources is one aspect, but this is meaningless if the various voices cannot be represented in the data structures used. Luckily, the Knowledge Graph as a data model and the Semantic Web as an information architecture provide excellent opportunities for maintaining various viewpoints on one subject. Its network structure and distributed nature is well-equipped to deal with such viewpoints. One example is found in Europeana, where the Europeana Data Model [10] allows for multiple publishers to provide information about one cultural heritage object. Building on the OAI-ORE model[4], each data provider provides a "Proxy" resource that

---

[4] http://www.openarchives.org/ore/1.0/datamodel

represents that object in the context of that provider and all metadata is attached to that Proxy resource, rather than the object representation itself. This allows to have multiple perspectives on a single real-world object. This model has also been applied to represent multiple and potentially disagreeing biographical descriptions of persons[16]. In the NewsReader project,[5] information on events was mined from newspaper articles. Multiple reports on the same event were identified via in- and cross-document coreference resolution and presented using named graphs to group statements about a particular event. Provenance information is then attached to these named graphs to represent the viewpoints of various providers[17, 7].

Where a voice corresponds to the view of a data publishers, such models appear adequate. At the other end of the spectrum, we see provenance at the level of the individual annotator or even the individual statement. WikiData for example, represents each statement as a tuple which includes the metainformation about that statement[21]. This allows for the recording of very precise provenance of statements, to the individual level.

The view of a data provider and the view of the individual annotator represent two points on the spectrum that can inform how to represent the more collective and elusive social-cultural voice. Here, we need models and design patterns on how voices can be represented. This means that not only "what is expressed" should be represented, but also "which worldview does this represent". One direction is that of "data lenses" as deployed for example in the OpenPhacts project[3]. Here sets of statements are annotated with provenance information, which in turn represent the scientific worldview of different types of end-users. In our case, we would like such lenses to correspond to various voices regardless of the end-user.

### 3.3 Usage of polyvocal knowledge

Representing and maintaining polyvocal knowledge is meaningless if it cannot be used in end-user facing applications. With the voices being represented in the knowledge graphs, different viewpoints are available for such applications. How to provide access to this more complex information in a meaningful way is a third major research challenge. The Europeana, Wikidata and GRaSP data models are already complex to query, even for Semantic Web experts. Adding another layer, that of the voice or lenses, potentially further complicates this.

Personalisation can be supported by these different views. Developers of end-user applications should take care that such personalisation does not lead to 'filter bubbles'. Effective communication of the various views that exist on specific resources is crucial. One example is objects that are assigned to different categories depending on the cultural context. Many objects in ethnography museums are currently categorised based on older (colonial) classification schemes, that might not represent currently held views, and moreover, such views are likely to

---

[5] https://newsreader-project.eu

differ per culture. If such polyvocal information about the object is maintained, it can be shown to the users in digital interfaces as well as in physical locations.

What is needed are design guidelines and patterns for visualisation of polyvocal Knowledge as well as reusable tools and methods.

## 4    Discussion

Creating a polyvocal and contextualised Semantic Web will be a community effort driven by interdisciplinary teams. Whilst the Cultural AI consortium is embedded in different cultural heritage institutions in the Netherlands that represent different voices, we have to be aware of the danger that this remains a singular (Western/Northern) voice. To create globally relevant and culturally aware semantic web resources, diversity and inclusivity is key, and for this we can for example take cues from Black in AI[6] and Widening NLP.[7] The Semantic Web community is a diverse one, as our conference attendants, organising and programme committees, editorial boards and mailing lists show. Different voices are present, so let us make our resources, models and projects reflect that.

## Acknowledgements

This paper builds upon the core concepts and research agenda of the Cultural AI Lab (https://www.cultural-ai.nl/), the authors would like to thank the co-creators of this research agenda: Antal van den Bosch, Laura Hollink, Martijn Kleppe, Johan Oomen, Jacco van Ossenbruggen, Stephan Raaijmakers, Saskia Scheltjens, Rosemarie van der Veen-Oei, and Lotte Wilms. We also want to thank our colleagues of the "Culturally Aware AI" project funded by the Dutch Research Council's Responsible use of AI programme, from the SABIO project, funded by the Dutch Digital Heritage Network, and from "Pressing Matter: Ownership, Value and the Question of Colonial Heritage in Museums", funded through the National Science Agenda of the Dutch Research Council.

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
