# OpenReview forum: "A Polyvocal and Contextualised Semantic Web"
_eswc-conferences.org/ESWC/2021/Conference/Research_Track — ESWC 2021 Research_

### Official Review · AnonReviewer3 · 2020-12-29
**A Polyvocal and Contextualised Semantic Web**

**Rating:** 1
**Confidence:** 5
**Impact:** 4
**Design And Technical Quality:** 3

**Review:**

The paper "A Polyvocal and Contextualised Semantic Web" is a submission in the new "problems to solve before you die" track, which "focuses on hard, longstanding or paradigm-breaking open problems" and on "the identification or even creation of new research trends and topics in or related to the Semantic Web." (according to the ESWC 2021 CFP).

Already in the abstract the authors observe that "the frequentist bias of many statistical analysis methods that are used to create for example knowledge graphs" results in simplified representations of the world in which diverse perspectives are underrepresented because the majority vote or the most frequent mention of a certain entity is always the most prominent and most salient one, drowning all the other voices. According to the authors, this bias needs to be addressed both in the algorithms analysing data and also in the resulting representations.

The submission focuses upon the implications of the lack of what the authors call "polyvocality" and contextual knowledge in the semantic web.

In Section 2, the authors briefly describe what a "voice" is by presenting different "perspectives" on a sample topic, which they illustrate using different "views" (cultural view, historical view, social view). A specific topic can be interpreted along different axes or dimensions, depending on the corresponding view and context. From the different terms used (voice, perspective, view), the authors generalise to "polyvocality" but it's not really clear why they picked "voice/polyvocality" over "view" or "perspective". The terms are used by the authors more or less synonymously but they mean different things. In that regard, a more thoroughly motivated choice of the terms used/preferred would've been appropriate.

At the end of Section 2 the authors state "It is complicated to identify, represent and show users different perspectives on an event, organisation, opinion or object." but they don't provide any evidence or explain why/how this is complicated. For example, isn't it possible to just select a few dozen properties to represent different types of "voices", "perspectives" or "views" (including a controlled vocabulary for the respective values), to standardise the approach within the community and then simply to use them in concrete knowledge graphs?

In Section 3, the authors provide more details and examples regarding the three challenges "identification", "representation" and "usage" of polyvocality. The authors also state that they present the "current state of the art" but they restrict themselves to Semantic Web research, i.e., they ignore all the relevant research made in more traditional Knowledge Representation research in the 70s, 80s and 90s. While Section 3 provides a good overview of the three different challenges picked by the authors, my suggestion would be to swap Section 3.2 and 3.1, simply because I think that representation needs to be addressed before one can address identification. In the Section on "usage", it's not really clear why the authors picked a singular ("another layer") in the sentence "Adding another layer, that of the voice or lenses, potentially further complicates this." Does adding a "voice" layer (singular) really only require adding one single layer to a knowledge graph? Is it really that simple?

Update: I read the rebuttals (thank you!). My overall score/evaluation remains unchanged.

**Anonymity:**

No, I would like my review to be deanonymized.

**Reuse And Availability:**

3: Medium

**Strong Points:**

Very relevant and interesting problem and overall challenge that clearly needs more attention in and from the Semantic Web community in the next couple of years.

**Subreviewer:**

I submitted this review.

**Weak Points:**

The terminology used/favoured by the authors is not well motivated and also should be re-evaluated (is it really "voice"/"polyvocality" that is the key missing piece or is it something like "perspective"/"multi-perspectivism"?).

The scope of the technical approach that needs to be worked on and finally devised needs to be described in more detail: what exactly needs to be developed? In that regard, what is a "bias or voice-aware information extraction method"?

With regard to the CFP of the "problems to solve before you die" track, the paper could, all in all, be even more ambitious in terms of scope of the suggested endeavour.

Larger parts of Section 3 and Section 4 seem to have been written in a bit of a hurry (blanks missing before citations, weird punctuation in Section 4). All of this should be fixed accordingly if the paper is accepted by the programme committee.

---

> ### Author Rebuttal · Authors · 2021-01-28
>
> We thank the reviewer for their constructive feedback and comments. We agree that in the current paper, the terms ‘view’, ‘perspective’ and ‘voice’ are used as near-synonyms. We can clarify our decision for the use of voice / polyvocality as the concept of focus more clearly by discussing the relations between the terms should the paper get accepted.
>
> The reviewer furthermore states that it is not clear why allowing for polyvocality is ‘difficult’ and asks “isn't it possible to just select a few dozen properties to represent different types of "voices", "perspectives" or "views"”. In the conversations with colleagues from humanities and cultural heritage institutions, we have found that the dimensions of the problem are as yet unknown; there are ‘known’ perspectives such as a (post)colonial perspective, or gender-biased perspectives, but often distinctions are not clear cut, and exist on a spectrum. Furthermore each of these also intersects with the type of data that is used as input and the processes performed on this data, so perhaps properties can be defined, and perhaps the solution should be sought in more fuzzy methods such as clustering. This is why we deem this a problem worthy of the ‘problems to solve before you die’ track.
>
> Regarding the single layer, this is a typo and should be layers, thanks for pointing this out!
>
> Regarding the suggestion to be even more ambitious in terms of scope of the suggested endeavour. We note in the paper that we investigate polyvocality and context from a cultural heritage perspective but that it impacts other domains too (we suspect any domain, although some more than others). In our answer to Reviewer 1, we suggest additional examples, that may mitigate this reviewer’s comment on our ambition.
>
> The reviewer notes that we “ ignore all the relevant research made in more traditional Knowledge Representation research in the 70s, 80s and 90s.”
> We mostly focused on current developments in Semantic Web, but of course methods proposed in the knowledge representation domain are highly relevant. We would appreciate it if the reviewer could provide us with suggestions to relevant prior work.

---

### Official Review · AnonReviewer2 · 2021-01-14
**Towards a Plural Semantic Web**

**Rating:** 2
**Confidence:** 4
**Impact:** 4
**Design And Technical Quality:** 4

**Review:**

This submission to the "Problems to Solve Before You Die" subtrack makes a convincing case for the need to adapt the tools and standard of the semantic Web to explicitly represent plural "voices", i.e., points of view, which means enabling the agents that use the knowledge stored in the semantic Web to exploit explicit meta-data about bias, context, etc., when performing inferences or presenting data to a user.
The analysis of what has been done so far and could be used to tackle this problem is rather complete, but I could not find any mentions of formalisms or tools to deal with uncertainty and/or imprecision, which certainly has a relation with the problem of polyvocality.

**Anonymity:**

Yes, I would like my review to remain anonymous.

**Reuse And Availability:**

3: Medium

**Strong Points:**

Good argument for why this problem deserves to be investigated.
Decent survey of what has been done so far.

**Subreviewer:**

I submitted this review.

**Weak Points:**

A discussion of how polyvocality relates to uncertainty and imprecision would add further interest and depth to the paper.

---

> ### Author Rebuttal · Authors · 2021-01-28
>
> We thank the reviewer for the kind and supportive review. We agree that there is a relation between polyvocality and (un)certainty. However, (un)certainty and imprecision play a role in any AI system and Semantic Web application and are being extensively researched so we did not discuss them specifically in our submission. Especially in identification, it is very likely that methods for this task will result in some (un)certainty about the outcome and methods that deal with polyvocality should be able to deal with this.

---

### Official Review · AnonReviewer4 · 2021-01-14
**An interesting perspective on an important problem**

**Rating:** 2
**Confidence:** 3
**Impact:** 4
**Design And Technical Quality:** 3

**Review:**

The authors point out the lack of support of multiple points of view in knowledge representations in general, and the Semantic Web in particular. They also warn against the dangers this represents in our society increasingly relying on AI-based systems. Finally, they the challenges such support would raise along the lifecycle of data, from collection to usage.

The paper is pleasant to read, and the problem addressed is indeed an important one. The authors introduce the concept of "voices" to capture this multiplicity of perspective – a concept that I find more interesting than the vaguer and over-used notion of "context".

Other minor remarks:

* p1: "current  AI  technologies  (...) often reflect the popular or majority vote": this is a rather restricted view of AI. I would rather have "AI" replaced here by "machine learning" or "data mining"
* please be consistent in the spelling of "Semantic Web" (with or without capital S and W))
* §4: the comma (",") between "voice" and "To" must be replaced with a period (".")

Meta-remarks:

* the 'Reuse And Availability' field below does not seem relevant for this subtrack (Problems to Solve Before You Die), so I picked the median rating by default

**Anonymity:**

No, I would like my review to be deanonymized.

**Reuse And Availability:**

3: Medium

**Strong Points:**

I like the fact that the authors do not only call for new models and formalisms, but also "design guidelines and patterns". The problem is not merely technical, but also cultural (!). Solving it will create challenges for developers *and end-users*, and I also like that the authors dedicated a subsection to usages of such polyvocal knowledge representations.


**Subreviewer:**

I submitted this review.

**Weak Points:**

While the addressed problem is definitely an important and a timely one, it has already been largely studied and described, as the authors themselves acknowledge. The concept of "voice", however, is an interesting (and, as far as I know, original in our field) way to look at it.

Another possible weakness of this paper is that it stays at a very high level, with little technical details on the proposed research directions.

---

> ### Author Rebuttal · Authors · 2021-01-28
>
> We would like to thank the reviewer for their review of our paper and appreciate the positive and supportive comments. Regarding the high-level description: indeed, for this specific “problems to solve before you die“ track, we opted to sketch the high-level issues rather than more detailed descriptions. In various projects related to the Cultural AI lab we are currently addressing these challenges in concrete domains and expect that more detailed descriptions of our research will appear as follow-up papers in subsequent ESWC proceedings. Furthermore, we hope that our work inspires other researchers in the Semantic Web and Knowledge Representation community to work on this topic.

---

### Official Review · AnonReviewer5 · 2021-01-15
**Interesting problem, worth discussing, but we might separate contextualization from polyvocality**

**Rating:** 1
**Confidence:** 4
**Impact:** 3
**Design And Technical Quality:** 4

**Review:**

The paper puts the focus on the problem of the representation of polyvocality of Semantic Web resources and presents three directions (identification, representation and usage) in which this issue should be investigated.

The paper highlights two relevant problems in the Semantic Web resources: moreover, the directions that are outlined are interesting and can contribute to the discussion in the ESWC community.

The work however concentrates mostly on the "polyvocality" side of the problem: context representation is, of course, a wider problem; on the other hand, while several technical solutions have been proposed in the last decades, a wider discussion on the directions for contextualization is in order.
In other words, I suggests to either concentrate on the polyvocality aspect of the contextualization problem or to expand the discussion to this wider problem.

About the description of the research directions, with respect to the "identification" challenge one might also to consider the dual problem of "coreference", i.e. recognize that, for example, a news text refers to a particular view of an event.
Related to identification is also the problem of the completeness of the views, i.e. recognize that all of the views of an event are fairly represented. From the representation point of view, the different views might be seen as "modifiers" of an unbiased default view of a resource.

The identified challenges for polyvocality are clearly described, however I suggest to add (where possible) more references to works that are already tackling some aspects of these challenges (e.g. representation of context or recognition of different views in semantic resources).


**Anonymity:**

Yes, I would like my review to remain anonymous.

**Reuse And Availability:**

4: High

**Strong Points:**

- The problems highlighted by the paper are relevant for the discussion in the community, both from an applicative and technical point of view
- The outlined challenges provide lines for further discussion of polyvocality aspects


**Subreviewer:**

I submitted this review.

**Weak Points:**

- The proposed challenges might be supported by more evidence, like e.g. works that currently investigate some of these aspects or current challenging application scenarios
- The contextual aspect is only considered with respect to the polyvocality: thus, I either suggest to concentrate on polyvocality or extend the discussion of challenges in the more wide contextualization problem.

---

> ### Author Rebuttal · Authors · 2021-01-28
>
> We thank the reviewer for their valuable comments, specifically about the relation between context representation and polyvocality. The goal of our paper was to include context(ualisation) only to the extent to which it relates to polyvocality. We can clarify this in the paper by more explicitly offsetting it against other types or uses of context and existing knowledge representation approaches for context.
> We are grateful for the co-referencing example presented by the reviewer which is indeed a great example of polyvocality and varying views on the same “thing”, in this case an event. Should the paper get accepted, we think this would be a valuable addition to the paper.

---

### Official Review · AnonReviewer1 · 2021-01-16
**Interesting problem but argument could be made stronger**

**Rating:** 1
**Confidence:** 3
**Impact:** 3
**Design And Technical Quality:** 3

**Review:**

In this "Problems to Solve Before You Die" submission, the authors highlight (the lack of) polivocality and contextual knowledge in the Semantic Web as a research problem that the community should spend more focus on. Besides explaining the problem, the authors also outline a vision for tackling the problem. The authors employ examples from the cultural heritage domain.

Through a number of examples, the authors have illustrated the lack of polivocality and contextual knowledge quite clearly. I personally agree that this is a problem that needs solving. However, I think a stronger argument why this problem should be solved could be very useful. Specifically, beyond the need of "a Japanese visitor to the Rijksmuseum", the urgency of the problem could be more emphasized if the authors can also illustrate the danger that can happen if the problem is not addressed. I think that this is also strongly related to recent developments in AI concerning societal and ethical impacts of unrecognized biases in data.

The three major challenges (in identification, representation, and usage) relevant to the problem are on point. Of these three, is there no existing work related to usage challenges? Moreover, I think the analysis on what has been done so far on those challenges need a more explicit argument why existing solutions have not been able to solve those challenges.
For example, why is CrowdTruth not enough? Why do GRaSP model or the data lenses in OpenPhacts project not solve the challenges adequately?

Overal, I think this is an interesting research direction to pursue. But I think the argument for it could have been made stronger. Thus, to me the paper is borderline, though I won't mind if the paper is accepted.

**Anonymity:**

Yes, I would like my review to remain anonymous.

**Reuse And Availability:**

3: Medium

**Strong Points:**

- Problem is clearly formulated, albeit in a form of high-level description.
- Relevant existing works toward the problem have been quite well-presented.

**Subreviewer:**

I submitted this review.

**Weak Points:**

- The argument of the urgency of the problem needs to be more explicit.
- Analysis on why existing works does not yet lead to a solution to the problem is not sufficiently provided.

---

> ### Author Rebuttal · Authors · 2021-01-28
>
> We thank the reviewer for their review of our paper. We are happy to read the positive and supportive comments and appreciate the constructive criticism. We will use these to improve our paper, should it get accepted: Regarding the high-level description: indeed, for this specific “problems to solve before you die“ track, we opted to sketch the high-level issues rather than more detailed descriptions. We are currently in multiple projects related to the Cultural AI lab working on these challenges in various specific projects and expect that more detailed descriptions of our research will appear as follow-up papers in subsequent ESWC proceedings as well as to inspire other researchers in the Semantic Web and Knowledge Representation community to work on this topic.
>
> The reviewer commented that the paper does not contain a thorough analysis of why certain solutions are not sufficient. We see the described solutions (lenses, crowdtruth-like solutions) definitely as pieces of the solution, but how these relate to an overall model of polyvocality is in our view an open issue. Regarding the urgency, we agree that this can be sharper in the direction suggested by the reviewer. Should the paper get accepted, we can add one or more additional examples on how unawareness of cultural context in either the data preparation or results interpretation can lead to undesired results. Here, we can current work on bias and fairness in Machine Learning or chatbots that quickly become racist would be relevant.

---

### Decision · Program_Chairs · 2021-02-23

**Decision:**

Accept

**Comment:**

All reviewers agree in their opinion that the paper raises an interesting and important idea, which is presented appropriately. Furthermore, all reviewers voted to accept this paper. However, for the final version of the paper, please also consider the cues mentioned by the reviewers regarding already existing work using different terminology.